# The Multifaceted Manifestations of Multisystem Inflammatory Syndrome during the SARS-CoV-2 Pandemic

**DOI:** 10.3390/pathogens11050556

**Published:** 2022-05-08

**Authors:** Héctor Raúl Pérez-Gómez, Rayo Morfín-Otero, Esteban González-Díaz, Sergio Esparza-Ahumada, Gerardo León-Garnica, Eduardo Rodríguez-Noriega

**Affiliations:** 1Centro Universitario Ciencias de la Salud, Universidad de Guadalajara, Guadalajara 44340, Jalisco, Mexico; hrulito@hotmail.com (H.R.P.-G.); rayomorfin@gmail.com (R.M.-O.); doc.glzdiaz@gmail.com (E.G.-D.); checo.esparza@gmail.com (S.E.-A.); madero247@gmail.com (G.L.-G.); 2Instituto de Patología Infecciosa y Experimental, Centro Universitario Ciencias de la Salud, Universidad de Guadalajara, Guadalajara 44280, Jalisco, Mexico; 3Hospital Civil de Guadalajara, Fray Antonio Alcalde, Guadalajara 44280, Jalisco, Mexico

**Keywords:** SARS-CoV-2, COVID-19, coronavirus, multisystem inflammatory syndrome, MIS-C, MIS-A, PASC, cytokine, chemokine abnormalities

## Abstract

The novel coronavirus SARS-CoV-2, which has similarities to the 2002–2003 severe acute respiratory syndrome coronavirus known as SARS-CoV-1, causes the infectious disease designated COVID-19 by the World Health Organization (Coronavirus Disease 2019). Although the first reports indicated that activity of the virus is centered in the lungs, it was soon acknowledged that SARS-CoV-2 causes a multisystem disease. Indeed, this new pathogen causes a variety of syndromes, including asymptomatic disease; mild disease; moderate disease; a severe form that requires hospitalization, intensive care, and mechanical ventilation; multisystem inflammatory disease; and a condition called long COVID or postacute sequelae of SARS-CoV-2 infection. Some of these syndromes resemble previously described disorders, including those with no confirmed etiology, such as Kawasaki disease. After recognition of a distinct multisystem inflammatory syndrome in children, followed by a similar syndrome in adults, various multisystem syndromes occurring during the pandemic associated or related to SARS-CoV-2 began to be identified. A typical pattern of cytokine and chemokine dysregulation occurs in these complex syndromes; however, the disorders have distinct immunological determinants that may help to differentiate them. This review discusses the origins of the different trajectories of the inflammatory syndromes related to SARS-CoV-2 infection.

## 1. Introduction

Since the discovery in 2019 of a new coronavirus, namely, severe acute respiratory syndrome coronavirus 2 (SARS-CoV-2), there has been an ongoing pandemic of coronavirus disease 2019 (COVID-19) [1,2]. Multiple organs are affected during SARS-CoV-2 infection. As a result, there is a persistent demand for information regarding how to diagnose the disease early, treat it promptly, and prevent it appropriately. Nevertheless, after 2 years of committed research, many questions remain, particularly with respect to how the pathogen affects the immune system such that it becomes a centerpiece of the disease process. The presence of the receptor for SARS-CoV-2, angiotensin-converting enzyme 2 (ACE2), in the vascular system and lung tissues creates the perfect conditions for thrombosis and inflammation [3,4,5,6].

After initial damage, with the acute phase manifesting as respiratory symptoms [7], an evolving multisystem inflammatory reaction is responsible for the multisystem inflammatory syndrome (MIS) seen in children (MIS-C) and adults (MIS-A), and in survivors with persistent symptoms.

It is thought that this development of immune system abnormalities occurs via a phenomenon similar to that occurring after the acute phase of other infectious diseases, indicating a postinfectious syndrome [8]. Therefore, infection of SARS-CoV-2, which causes COVID-19 by inducing an imbalance in the immune system with abnormal innate and adaptive immune responses and inflammasome activation in various populations, leads to severe multisystem inflammatory syndromes.

This review analyzes the various manifestations of infection with the original SARS-CoV-2 virus and its variants. The review focuses on clinical manifestations and their relationships to the diverse immune responses in MIS developing after or during COVID-19.

## 2. Coronavirus Infection

Coronavirus infection in humans primarily affects the respiratory tract, usually causing a self-limiting disease (such as a common cold) or, rarely, moderate disease (such as bronchitis or pneumonia) [9]. Coronaviruses also cause disease in a variety of animals, including gastroenteritis in bovines, cats, dogs, and turkeys [9]. Coronaviruses 229E, OC43, NL63, and HKU1, also known as seasonal coronaviruses, are responsible for common cold-like syndromes (229E, OC43) [10,11], and these four types of seasonal coronaviruses cause disease and outbreaks during winter [12].

In 2002, an aggressive coronavirus was discovered in China, causing fever, myalgia, headache, chills, diarrhea, nonproductive cough, dyspnea, and pulmonary infiltrates. Due to this presentation, the virus was called severe acute respiratory syndrome virus (SARS-CoV) [13]. In 2012, another very pathogenic coronavirus was identified in Saudi Arabia [14] and was termed Middle East respiratory syndrome coronavirus (MERS-CoV) [15]. MERS-CoV can produce severe pneumonia with multisystem involvement [16]. MERS begins with fever, cough, sore throat, myalgias, arthralgias, vomiting, and diarrhea. This is followed by dyspnea, pneumonia, and multisystem involvement, predominantly including renal damage [16]. For MERS, people with comorbidities have the highest morbidity and mortality rates [17].

In 2019, the newest aggressive coronavirus was discovered in Guangdong, China. Due to similarities with the original SARS virus, it was designated SARS-CoV-2 [1,2], and it primarily affects the lungs. However, shortly after the first descriptions, the virus was shown to also attack multiple organs and impact people > 65 years of age as well as those with diverse comorbidities [18,19,20,21]. The virus rapidly disseminated worldwide to create a pandemic of a disease now known as coronavirus disease 2019 (COVID-19).

As previously mentioned, seasonal coronavirus usually causes a self-limiting disease resembling a common cold. In contrast, the coronaviruses SARS-CoV, MERS, and SARS-CoV-2 are responsible for severe syndromes. In the first report of SARS-CoV infections, other findings beyond the usual presentation of atypical community-acquired pneumonia—including diarrhea and a maculopapular rash, which are both manifestations of multisystem involvement were described [22]. For MERS, vomiting (21%), diarrhea (26%), and renal damage also demonstrate multisystem involvement; in addition, damage is more frequent in older patients (>65 years), immunocompromised patients, and patients with other comorbidities, including obesity, diabetes, chronic cardiovascular disease, and chronic lung disease [16].

SARS-CoV-2, with a greater affinity for the receptor ACE-2, which is prominently found in endothelial cells in the inner lining of all blood vessels, leads to inflammation and clotting, involving a cascade of events responsible for damage to the lungs, respiratory system, heart, kidneys, gastrointestinal tract, adipose tissue, skin, and brain [4,23].

In the first clinical description of infected patients in early 2020, fever was present in 98%, with cough seen in 76%, dyspnea in 55%, and myalgia or fatigue in 44% [24]. Soon after the initial clinical descriptions, it was evident that there is a remarkable diversity in COVID-19 and its clinical presentation.

In 2021, investigators described six distinct symptom clusters of presentation occurring in the first 5 days of illness, as based on symptoms logged into a COVID symptom smartphone application [25].

In symptom cluster one (flu-like with no fever), headache, loss of smell, muscle pains, cough, sore throat, and chest pain were noted. For symptom cluster two (flu-like with fever), the symptoms described for cluster one in addition to hoarseness and loss of appetite were reported. Gastrointestinal cluster diarrhea was included in symptom cluster three. Severe level one fatigue was added for symptom cluster four and severe level two confusion was evident with symptom cluster five. Lastly, severe level three abdominal pain, shortness of breath, and abdominal pain were included for symptom cluster six [25].

Moreover, diverse viral strains can modify the epidemiology or pathogenesis of COVID-19 [26].

During the first 28 months of the pandemic, various mutations occurred in the virus, resulting in some variants having increased transmissibility and causing more severe disease. For example, the Delta variant became more distinctive than the original Wuhan strain (wildtype), Alpha, Beta, Gamma, Delta, and Omicron, causing more severe cases with more hospitalizations and deaths.

Indeed, the Delta variant was the predominant strain worldwide until the Omicron variant appeared and replaced it to become the leading strain in 2022. The Omicron variant is also highly transmissible and can evade the immunity induced by vaccination or natural infection, though it has decreased pathogenicity, producing less hospitalization, fewer deaths, and fewer severe cases [27].

Similar to the Delta variant, the Omicron variant will result in MIS-C in patients with distinctive risk factors for severe disease, including older age, fever, obesity, and seizures, prompting hospitalization and intensive care [28,29].

## 3. Multisystem Inflammatory Syndrome in Children (MIS-C)

The term multisystem inflammatory syndrome was initially introduced to describe children affected by a new syndrome that emerges after COVID-19. In April 2020, a cluster of eight children with hyperinflammatory shock in London, U.K., was described [30]. The patients exhibited other findings similar to Kawasaki disease (KD) or toxic shock syndrome (TSS). The clinical presentation included fever (>38 °C), rash, conjunctivitis, peripheral edema, pain in the extremities, abdominal pain, vomiting, and diarrhea. There were laboratory markers of inflammation, including elevated C-reactive protein, procalcitonin, ferritin, and D-dimer. Four children had family exposure to SARS-CoV-2, and two others were exposed after discharge. The authors suggested that the phenomenon is related to previous asymptomatic SARS-CoV-2 infection [30].

Additionally, this report was followed by the description of 58 children from eight hospitals in England admitted between 23 March and 16 May 2020, who showed persistent fever and laboratory evidence of inflammation, meeting the published definitions for pediatric inflammatory multisystem syndrome (PIMS-TS) temporally associated with SARS-CoV-2 [31]. The patients’ characteristics were compared with the clinical characteristics of patients with KD, KD shock syndrome, and TSS; 78% had evidence of current or prior SARS-CoV-2 infection. All children presented with fever (100%), vomiting (45%), abdominal pain (53%), and diarrhea (52%). Rash was present in 52% of cases and conjunctival injection in 45%. Comparison of PIMS-TS with KD and with KD shock syndrome has revealed differences, including older age (median: 9 years vs. 2.7 years and 3.8 years, respectively). Furthermore, laboratory evaluation results were consistent with marked inflammation, and eight patients (14%) developed coronary artery dilatation or aneurysm [31]. In April 2020, a cluster of 78 children with PIMS-TS required admission to a pediatric intensive care unit [32]. The increase in this new syndrome compelled investigators to compare it with four similar inflammatory conditions: KD, TSS, hemophagocytic lymphohistiocytosis, and macrophage activation syndrome [32]. The median age of the patients was 11 years, and 67% were male. Presentations included fever (100%), shock (87%), abdominal pain (62%), vomiting (63%), and diarrhea (64%). During treatment, 46% of the patients were invasively ventilated, 36% had evidence of coronary artery abnormalities, three needed extracorporeal membrane oxygenation, and two died [32].

This new MIS-C was also encountered in the U.S. between March and May 2020. In 2020, 186 cases were reported from 20 states in the U.S. following a case definition that included severe illness requiring hospitalization, age < 21 years, fever for at least 24 h, laboratory evidence of inflammation, multisystem involvement, and confirmed infection with SARS-CoV-2 [33]. The median age was 8.3 years, and 62% were males. Laboratory findings included elevated C-protein, lymphocytopenia, elevated ferritin, D-dimer, elevated aspartate transaminase (AST), anemia, high fibrinogen, and elevated troponin. Bodily systems involved included the gastrointestinal tract in >90% of cases, with hepatomegaly, hepatitis, pancreatitis, deep vein thrombosis, pulmonary embolism, mucocutaneous involvement, arthritis, arthralgias, myositis, myalgias, neurologic involvement, and acute kidney injury. Cardiovascular involvement was found in 80% of cases, with pericarditis, pericardial effusion, arrhythmia, and elevated levels of brain natriuretic peptide [33].

During the same time frame, mid-2020, 101 patients with MIS-C were described in the state of New York [34]. Thirty-one percent of the patients were 0–5 years of age, 42% were 6–12 years, and 20% were 13–20 years of age. Fifty-four percent were males, 97% presented with tachycardia, 80% with gastrointestinal symptoms, 60% with rash, 56% with conjunctival injection, and 27% with mucosal changes. All had a temporal relationship with the COVID-19 outbreak in the state of New York. The cases in this series were divided into five clusters: dermatologic or mucocutaneous, gastrointestinal, KD or atypical KD, myocarditis, and neurologic. The dermatologic or mucocutaneous cluster was seen more frequently in the 0–5 age group, the gastrointestinal in the 6–16 age group, KD in the 0–5 age group, myocarditis in the 13–20 age group, and neurologic symptoms in the 6–12 and 13–20 age groups [34].

The Centers for Disease Control and Prevention (CDC, Atlanta, GA, USA) reported that by 29 July 2020, 570 MIS-C cases had been diagnosed in the U.S. Two hundred and three (35.6%) of the patients had a clinical course consistent with previously published MIS-C reports, as characterized predominantly by shock, cardiac dysfunction, abdominal pain, and markedly elevated inflammatory markers, and almost all had positive SARS-CoV-2 test results [35]. The remaining 64.4% of MIS-C patients displayed manifestations that appeared to overlap with acute COVID-19, had a less severe clinical course, or had features of KD. The median duration of hospitalization was 6 days; 364 patients (63.9%) required care in an intensive care unit, and 10 (1.8%) died [35]. Up to June 2020, the incidence of MIS-C per 1,000,000 person-months was 5.1 persons (95% CI, 4.5–5.8) in the U.S. [36].

The heart is distinctively involved in MIS-C, with a 100% incidence of left ventricular dysfunction in MIS-C, 17% incidence of coronary dilatation, and 8% incidence of pericarditis. Such cardiac injury can result in acute heart failure [37]. Through evaluation of 124 children hospitalized with COVID-19, 63 were found to have MIS-C, and cardiac complications were more frequent in critically ill patients with MIS-C (55% vs. 28%; *p* = 0.04), including systolic myocardial dysfunction (39% vs. 3%; *p* = 0.001) and valvular regurgitation (33% vs. 7%; *p* = 0.01) [38].

Using echocardiography and cardiac magnetic resonance imaging, 60 controls and 60 cases of multisystem inflammatory syndrome in children were evaluated for cardiac outcomes at 3–4 months after initial presentation [39]. All deformation parameters, including left ventricular global longitudinal strain, peak left atrial pressure, longitudinal early diastolic strain rate, and right ventricular free wall strain, recovered quickly within the first week, which was followed by continued improvement and complete normalization at 3 months [39].

The development of a multisystem inflammatory syndrome in children with various cardiac abnormalities during the COVID-19 pandemic has compelled comparisons with KD. Although first described in 1974, the etiology of KD is still unknown [40]. Initially, KS was described as a mucocutaneous lymph-node syndrome, and the original report showed the most prominent findings in children [41,42]. Since that description, KD has become known as a multisystem disease affecting multiple organs, including the heart, skin, pancreas, and liver [43,44]. A recent study analyzed 233 MIS-C patients: 102 with COVID-19, 101 with KD, and 76 with TSS [45]. Patients with MIS-C more frequently exhibited decreased cardiac function (38.6%), myocarditis (34.3%), pericardial effusion (38.2%), mitral regurgitation (31.8%), and pleural effusion (34.8%) compared to patients with the other conditions [45]. In addition, patients with MIS-C had increased peak C-reactive protein levels and decreased platelets and lymphocyte nadir counts compared with patients with COVID-19 and KD. Those with MIS-C also showed elevated levels of troponin, brain natriuretic peptide, and pro-brain natriuretic peptide compared to those with COVID-19 [45]. The authors proposed a series of diagnostic scores for distinguishing MIS-C from COVID-19, KD, and TSS for timely and accurate diagnoses [45]. The different abnormalities in MIS-C and KD as well as shared abnormalities are presented in Figure 1.

Children with COVID-19 and MIS-C may develop a variety of neurologic symptoms, including severe encephalopathy, stroke, central nervous system infection or demyelination, Guillain–Barre syndrome or variants, and acute fulminant cerebral edema. Patients with neurologic involvement are more likely to have underlying neurologic disorders. Of the 43 patients who developed COVID-19-related life-threatening neurologic involvement in one study, 17 survivors (40%) had new neurologic deficits at hospital discharge, and 11 patients (26%) died [46].

Based on multiple reviews of MIS-C, there is agreement that this emerging entity is a pediatric disease with a high incidence of gastrointestinal symptoms. It is dangerous and potentially lethal, with some patients requiring critical care support [47]. Measurement of inflammatory markers assists in clinical evaluation [48]. Although most children survive with prompt recognition and medical attention, the long-term outcomes of this condition remain unknown [49]. Fortunately, vaccination appears to prevent MIS-C [50].

The abnormalities in MIS-C and COVID-19 as well as shared anomalies are provided in Figure 2.

## 4. Multisystem Inflammatory Syndrome in Adults (MIS-A)

Multisystem inflammatory syndrome can also occur in adult patients, which is termed MIS-A.

In one of the first cases reported, a 45-year-old male who cared for his wife with COVID-19 developed fever, sore throat, diarrhea, lower extremity pain, and conjunctivitis before a diagnosis of MIS-A KD was considered [51].

One study in adults identified MIS-A in 16 patients: 9 were reported to the CDC, and 7 were case reports published previously in the U.S. and the U.K. [52]. Age ranged from 21 to 50 years; seven of the patients were males. Nine had no underlying medical condition, six had obesity, one had diabetes mellitus, two had hypertension, and one had obstructive sleep apnea. Eight patients had a history of previous respiratory illness; eight did not. Fever lasting >24 h was documented in 12 cases; 6 had chest pain or palpitations, and all had cardiac involvement, an abnormal electrocardiogram, arrhythmias, and elevated troponin levels [52].

In an analysis of 7196 patients with SARS-CoV-2 infection, 156 patients (11.7%) were classified as being at risk for MIS-A; further investigation revealed that 15 of these 156 patients (9.6%) met the criteria for MIS-A [53]. Patients with MIS-A were younger (median age 45.1 years vs. 56.5 years for patients admitted for acute COVID-19 symptoms) and more likely to have evidence of SARS-CoV-2 infection documented by serologic testing (60.0% with MIS-A vs. none with COVID-19; *p* < 0.001). Nine of the 15 patients with MIS-A (60%) had acute COVID-19 symptoms, and 3 (20%) required admission for acute COVID-19 before being admitted for MIS-A. Among patients with prior admission for acute COVID-19, the median interval between acute COVID-19 admission and MIS-A admission was 23 days. The median number of organ systems involved was four, with the gastrointestinal, hematologic, and kidney systems being most commonly affected [53].

In one systematic review of 221 patients with MIS-A, the median age was 21 years, 70% were men, and 58% had no underlying comorbidity [54]. Sixty-eight percent noted a previous symptomatic COVID-19-like illness. Most patients with MIS-A presented with fever (96%), hypotension (60%), cardiac dysfunction (54%), shortness of breath (52%), and diarrhea (52%), and the median number of organ systems involved was five. The median hospital stay was 8 days; 57% were admitted to the intensive care unit, 47% required respiratory support, and 7% died [54].

## 5. MIS-A and MIS-C after Vaccination

Although unusual, MIS-A can develop after COVID-19 vaccination. One question is whether hyperinflammation due to antibody-dependent enhancement can follow COVID-19 or vaccination against SARS-CoV-2.

In one of the first reports, 20 patients who met the case definition for MIS-A were reported to the CDC. Their median age was 35 years (21–66 years), and 13 (65%) were male. Overall, 16 (80%) patients had a preceding COVID-19-like illness, with a median of 26 days (range 11–78 days), before MIS-A onset. Additionally, all 20 patients had laboratory evidence of SARS-CoV-2 infection. Seven MIS-A patients (35%) had received the COVID-19 vaccine a median of 10 days (range, 6–45 days) before MIS-A onset; three patients had received a second dose of the COVID-19 vaccine at 4, 17, and 22 days before MIS-A onset. Patients with MIS-A predominantly had gastrointestinal and cardiac manifestations and hypotension or shock [55].

Several other case reports have demonstrated MIS after vaccination, including a report of three patients in California, U.S., who experienced MIS after immunization and SARS-CoV-2 infection [56].

A 44-year-old woman began experiencing pain at the site of her COVID-19 vaccination (Pfizer-BioNTech mRNA); fever, diarrhea, and abdominal pain ensued over the following days. On evaluation, an erythematous rash and subcutaneous edema were noted on the chest, and levels of C reactive protein and troponin were elevated [57]. During admission, she developed a pulmonary embolism and acute kidney injury; most of the noted abnormalities were resolved by treatment with methylprednisolone. The investigators suggested MIS after SARS-CoV-2 vaccination (MIS-V) [57].

Fatality after COVID-19 vaccination has occurred, albeit infrequently. One healthcare worker who was previously healthy became asymptomatic at 6 days after the onset of COVID-19 symptoms and received the first dose of Pfizer/BioNTech mRNA COVID-19 vaccine; he then received the second dose 20 days later. Twenty-two days after receiving the second dose of the COVID-19 vaccine, he developed fever, malaise, headache, and odynophagia [58]. The patient visited an emergency department 4 days later due to worsening symptoms. On examination, cervical lymphadenopathy, marked bilateral conjunctival erythema, and a faint papular rash on the pelvis and left flank were noted. Laboratory testing revealed thrombocytopenia, elevated levels of C-reactive protein at 284.0 mg/L, serum ferritin at 1434.9 ng/mL, and troponin-I at 18.0 ng/mL [58]. He was admitted to the hospital. On the third day of hospitalization, the patient had an acute change in mental status, including confusion and global aphasia. The patient died on their fourth day in hospital. Serum antibody results were negative for SARS-CoV-2 IgM but positive for IgG. Serum showed a high titer of anti-spike receptor-binding domain antibody (1:75,000) compared with a naturally infected SARS-CoV-2-positive control (1:4000). Notable findings on gross internal autopsy examination included 525 mL pericardial effusion, cardiac enlargement, 5 L hemoperitoneum, and a 20 cm diameter perisplenic hematoma. Microscopy analysis of the lungs showed diffuse congestion, increased intra-alveolar macrophages, multifocal hemorrhage, capillaritis, and disseminated microthrombi [58].

After vaccination for COVID-19 commenced in the pediatric population, reports of MIS-C began. These reports included a 12-year-old male who had no prior SARS-CoV-2 infection or exposure but developed MIS-C after his first dose of the COVID-19 mRNA vaccine [59].

An 18-year-old adolescent developed MIS-C at 10 weeks after a second vaccination with the SARS-CoV-2 vaccine from Pfizer/BionTech (BNT162b2), with a fever (up to 40 °C) [60]. The patient was hospitalized 14 days later, and pericardial effusion (10 mm) was diagnosed by echocardiography. C-reactive protein (174 mg/L), NT-BNP (280 pg/mL), and troponin T (28 pg/mL) levels were all elevated [60].

## 6. Postacute Sequelae of SARS-CoV-2 Infection

After an episode of acute COVID-19, numerous patients have displayed various persistent symptoms. This disorder has received multiple names, including long COVID, long haulers, long-term COVID, post-COVID syndrome, and postacute sequelae of SARS-CoV-2 infection (PASC). In follow-up observations at 60 days after an episode of COVID-19, 32% of patients had 1–2 symptoms, and 55% had >3 symptoms. The most frequent symptoms in this series were fatigue, dyspnea, and joint pain [61]. When severe fatigue was present in the acute episode, it persisted up to 6 months after the acute episode ended [62]. In another study, new-onset diabetes mellitus, major adverse cardiovascular events, chronic kidney disease, and chronic liver disease were found during the follow-up of patients with PASC [63].

In PASC, persisting abnormality in some laboratory and X-ray studies can continue until 60 days after the acute episode. In patients with persistent fatigue (69%), dyspnea (53%), cough (34%), and continuing anosmia at 60 days after the acute episode, 30.1% showed elevated D-dimer levels, and 38% had an abnormal chest X-ray [64].

In a prospective observational study that included 4182 cases of PASC, 13.3% had symptoms for ≥28 days, 4.5% for ≥8 weeks, and 2.3% for ≥12 weeks [65]. Fatigue, headache, dyspnea, and anosmia are more likely to persist in patients with older age or increased body mass index, in females, and when ≥5 symptoms occurred during the first week of illness [65].

Does PASC appear in the population affected by COVID-19, and if so, what is its prevalence?

In a sample of 593 patients (56.1% females), those reporting very severe symptoms had a 2.25 times higher prevalence of 30-day persistent COVID-19 and a 1.71 times higher prevalence of 60-day persistent COVID-19 than those with mild initial symptoms. Hospitalized patients had an approximately 40% higher prevalence of 30-day and 60-day persistent COVID-19 than nonhospitalized patients. The investigators concluded that PASC is highly prevalent among cases involving severe initial symptoms and, to a lesser extent, in patients reporting mild and moderate symptoms [66].

Are there more data on the PASC burden? In a cohort of 181,384 people with COVID-19 and 4,397,509 noninfected controls from the US Department of Veterans Affairs healthcare databases, PASC was defined as the presence of at least one sequela over noninfected controls. COVID-19 cases involved 73.43 cases of PASC per 1000 persons at 6 months [67].

Does PASC occur more frequently in females or males? A single-center study in Italy evaluated 377 patients and found that female sex was independently associated with PASC (AOR 3.3 vs. males, 95% CI 1.8–6.2, *p* < 0.0001), followed by advanced age (AOR 1.03 for 10 years older, 95% CI 1.01–1.05, *p* = 0.01) and active smoking (AOR 0.19 for former smokers vs. active smokers, 95% CI 0.06–0.62, *p* = 0.002) [68].

There is also the question of whether PASC occurs in nonhospitalized patients. In 303 nonhospitalized individuals with a positive COVID-19 test who were followed for a median of 61 days (range 30–250); the COVID-19-positive participants were mostly female (70%). The prevalence of PASC at 30 days postinfection was 68.7% (95% confidence interval 63.4–73.9), and the most common symptoms were fatigue (37.5%), shortness of breath (37.5%), brain fog (30.8%), and stress or anxiety (30.8%). The median number of symptoms was 3 (range 1–20). Among 157 participants with longer follow-up (>60 days), the PASC prevalence was 77.1% [69].

In an investigation (Northern Colorado SARS-CoV-2 Biorepository) that included adults from the community or previously hospitalized with a history of a positive polymerase chain reaction (PCR) test for SARS-CoV-2, 93% of hospitalized participants developed PASC. In contrast, 23% of those not requiring hospitalization developed PASC. At 90–174 days after the SARS-CoV-2 diagnosis, 67% of all participants had persistent symptoms (*n* = 37 of 55), and 85% percent of participants who required hospitalization during initial infection (*n* = 20) still had symptoms. The most common symptoms reported after 15 days of infection were fatigue, loss of smell, loss of taste, exercise intolerance, and cognitive dysfunction. The researchers concluded that patients hospitalized for COVID-19 are significantly more likely to have PASC than those not requiring hospitalization; nonetheless, 23% of patients who were not hospitalized also developed PASC [70].

PASC can have different manifestations in young patients, including myocarditis. In a case-control study investigating cardiovascular involvement in PASC (CV-PASC), 50 soldier patients and 50 healthy soldier controls were included; the median time from SARS-CoV-2 detection to evaluation was 71 days. Only 10% of the patients required hospitalization, and right ventricular ejection fraction was reduced in the cases compared to the controls (51.0% vs. 53.2%, *p* = 0.012). Four patients were diagnosed with myocarditis (8%) and 1 (2%) with Takotsubo cardiomyopathy; 1 (2%) had new biventricular systolic dysfunction of unclear etiology [71].

In a population-based, longitudinal, multicenter study in Sweden involving data for patients with MIS-C prospectively collected between 1 December 2020 and 31 May 2021, 36% had persistent symptoms at 8 weeks after MIS-C was diagnosed, and 5% had abnormal echocardiograms. Older age and higher levels of initial care appeared to be risk factors [72].

PASC in children (PASC-C) can have critical neurologic manifestations, especially after MIS-C. In a cohort of 47 patients, 77% reported neurological symptoms, 60% reported psychiatric symptoms, and 77% reported sleep symptoms during hospitalization. Prior to hospitalization, 15% reported neurological symptoms, 0% psychiatric symptoms, and 7% sleep symptoms; moreover, 18 (50%) of the 36 patients who had neurological symptoms during hospitalization continued to have symptoms on follow-up. Similarly, 16 (57%) of 28 patients with psychiatric symptoms reported persistence at follow-up. Fifteen (42%) of the 18 patients reporting sleep disturbance during hospitalization also had persistence during follow-up. The aggregate of neurological, psychiatric, and sleep symptoms during admission and at follow-up was significantly higher for MIS-C patients requiring intensive care than for the control group (*p* = 0.01).

In general, the factors related to the development of PASC-C and PASC-A appear to be identical. Factors leading to PASC include the following: effects from acute SARS-CoV-2 injury in one or multiple organs; persistent reservoirs of SARS-CoV-2; reactivation of neurotrophic pathogens, such as herpesviruses, due to COVID-19 causing immune dysregulation; ongoing activity of primed immune cells; autoimmunity due to molecular mimicry between pathogen and host proteins [73,74]; the appearance of antibodies specific to ACE2 [75]; and an elevated number of monocytes containing SARS-CoV-2 S1 protein in both severe COVID-19 and PASC patients [76]. Another feature is significant differences between innate (NK cells, LD neutrophils, and CXCR3+ monocytes) and adaptive (T helper, T follicular helper, and regulatory T cells) immune populations in convalescent individuals compared to healthy controls. This is most strongly evident at 12–16 weeks postinfection [77].

The characteristics of MIS-A and PASC as well as shared characteristics are given in Figure 3.

## 7. Discussion

The multiple presentations of SARS-CoV-2 infection, including MIS, make diagnosing COVID-19 a daunting differential process. Patients may present with mild, moderate, or severe COVID-19. Some patients may have no symptoms, yet may develop MIS-C or MIS-A weeks later.

COVID-19 can resemble influenza, other respiratory viruses, and acute community-acquired pneumonia in the pediatric population, and MIS-C must be differentiated from KD.

In adult patients, COVID-19 can resemble influenza, other respiratory viruses, and acute community-acquired pneumonia, similar to the pediatric population, but MIS-A should be separated from TSS.

Furthermore, new SARS-CoV-2 variants have emerged, complicating diagnosis and causing previous epidemiological concepts to change.

The appearance of the Delta variant made us aware of the possibility of a variant with increased transmissibility, transmitting faster in households with a decreased generation time.

Although this new variant did not modify the usual presentation, it did result in increased morbidity and mortality [78,79].

The Omicron variant, which carries 37 mutations in the spike protein and 15 amino acid substitutions in the receptor-binding domain, appears to preferentially infect and replicate in the upper respiratory tract. This is in comparison to Delta and other variants that favor the lower respiratory tract, and Omicron has a decreased ability to infect lung tissue [80].

The Omicron variant is a new variant similar to Delta with increased transmissibility. However, when the prevalence was >70%, it was noted that loss of smell is less common and sore throat more common, and that fewer patients required hospitalization [81].

In countries where it outcompeted Delta, Omicron showed the potential to evade the immunity induced by vaccination or natural infection. Regardless, secondary to its decreased pathogenicity, it produced fewer deaths and fewer excess deaths [27].

The Omicron variant has increased the hospitalizations of children aged 5–11 years compared to the Delta variant, though it is still unknown whether this increase in severe cases will cause a rise in MIS-C [29].

In differential diagnosis of MIS-A and PASC in the adult population, evaluations are complicated by secondary attacks of new variants. In addition, secondary infections have different presentations than the primary infection; patients are more frequently asymptomatic or have mild disease with symptoms that suggest PASC. There is growing evidence of newer presentations in the first episode of COVID-19 in adult patients; despite this, MIS-A often presents with fever, mucocutaneous lesions, and conjunctivitis (Figure 3). Overall, MIS-A runs its course in weeks, whereas the duration of PASC is measured in months.

PASC is now a concerning, long-lasting complication of COVID-19. The sequelae are more often seen in patients with a severe first episode requiring hospitalization. In some patients, some of the presenting symptoms of the initial attack, such as loss of smell or taste, continue.

It is calculated that PASC (long COVID, post-COVID-19 condition) has a global prevalence of 0.43%, higher in hospitalized patients (0.54%) and lower in nonhospitalized patients (0.34%) [82].

PASC at 30, 60, 90, and 120 days after development occurred in 0.37%, 0.25%, 0.32%, and 0.49% of cases, respectively [82].

PASC has distinct clinical phenotypes that help with diagnosis [83]. There are three different clusters: cluster one involves pain symptoms such as joint pain, myalgia, and headache; cluster two involves cardiovascular symptoms such as chest pain, shortness of breath, and palpitations predominate; and cluster three involves significantly fewer symptoms [83].

Recent investigations into how SARS-CoV-2 affects the brain and its long-term effects have reported that brain damage occurs in specific areas; for example, the brain areas related to the sense of smell are affected [84]. In addition, a study of people aged 51–81 years who underwent brain scans both before COVID-19 and several months later revealed damage to distinct brain areas and loss of gray matter greater than the annual loss [84].

This evolving, recently described, mechanism of brain damage may explain not only specific symptoms, such as loss of smell, but also other PASC features, such as brain fog and cognitive dysfunction. Normal cognitive function is required for normal daily activities throughout life. In a 1-year follow-up study of older COVID-19 survivors, four categories of cognitive function were considered: stable cognition, early-onset cognitive decline, late-onset cognitive decline, and progressive cognitive decline [85]. Previous severe COVID-19 was associated with a higher risk of early-onset cognitive decline, late-onset cognitive decline, and progressive cognitive decline.

As the world enters the third year of the COVID-19 pandemic, not only PASC but also other long-term consequences are being thoughtfully studied. These include new diagnoses of shortness of breath, heart-rate abnormalities, and type 2 diabetes, adding to the long list of complications after acute COVID-19 [86].

It is crucial to continue investigating all the probable mechanisms of inflammation and immune responses in COVID-19, MIS-C, MIS-A, and PASC.

Further studies should explain why pediatric patients with MIS-C have higher levels of SARS-CoV-2 spike IgG than children with severe COVID-19 [87]; excessive inflammatory response; profound lymphopenia; broken cytokine production, including interferon-gamma, interleukin-7, and interleukin-22; and T-cell exhaustion [88]. Moreover, how to use the phases of immune damage including the process of invasion, the primary blockade of innate antiviral immunity, engagement with the virus, and the immune changes responsible for acute and long-term complications should be explored [89]. Finally, the portion of the COVID-19 pathogenesis of antibody-mediated SARS-CoV-2 uptake by monocytes/macrophages that triggers inflammatory cell death, interrupting the production of the infectious virus but causing systemic inflammation, should be examined [90].

In addition, the role of mast cells in all syndromes, especially in PASC, should be further studied. SARS-CoV-2 activates mast cells, and the response leading to hyperinflammation is mediated by the release of histamine followed by proteases, cytokines, chemokines, prostaglandins, and leukotrienes [91,92]. In PASC, mast cell activation is increased, which is responsible for prolonged symptomatology [93].

The phenomenon of antibody-dependent enhancement should also be considered because it may play a role in the pathogenesis of all inflammatory syndromes occurring after COVID-19 and after vaccination. For example, SARS-CoV-2 antibodies bound to Fc receptors on macrophages and mast cells might be responsible for ADE in patients [94].

Two years have passed since the description of SARS-CoV-2; new variants have appeared, vaccines are providing help, some therapeutic modalities have been abandoned, new oral antivirals are becoming available, and new surveillance methodology is being discussed [95]. However, the multiple presentations of SARS-CoV-2 infection still present a challenge to the global scientific community.

## Figures and Tables

**Figure 1 pathogens-11-00556-f001:**
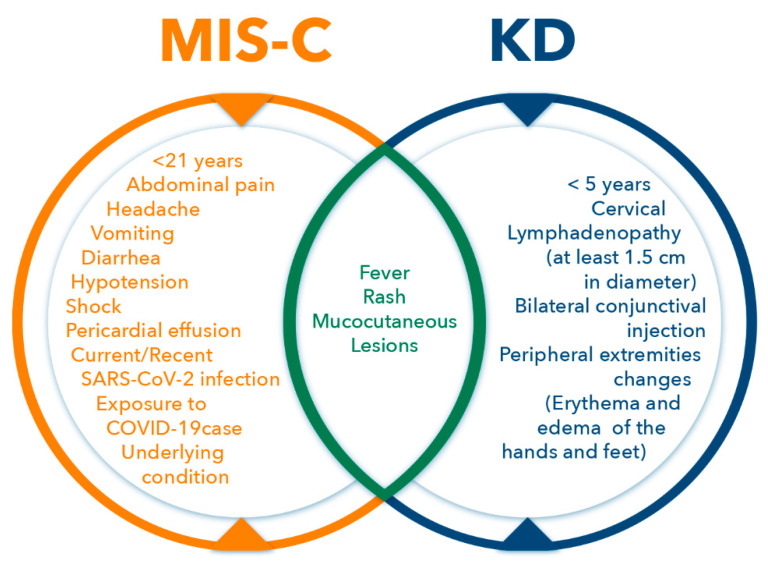
Venn diagram describing findings in Multisystem Inflammatory Syndrome in Children (MIS-C), Kawasaki Disease (KD), and shared abnormalities. Abbreviations: SARS-CoV-2—Severe Acute Respiratory Syndrome Coronavirus 2; COVID-19—Coronavirus Disease 2019.

**Figure 2 pathogens-11-00556-f002:**
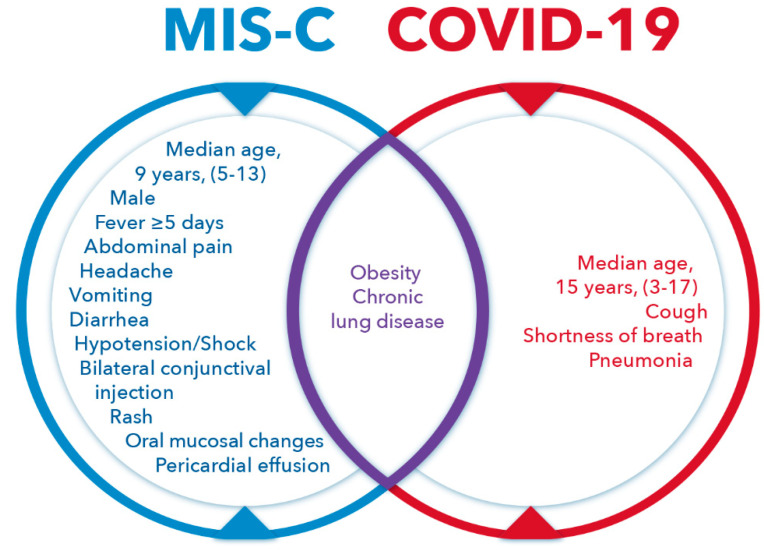
Venn diagram illustrating abnormalities in Multisystem Inflammatory Syndrome in Children (MIS-C), Coronavirus Disease 2019 (COVID-19), and joint abnormalities.

**Figure 3 pathogens-11-00556-f003:**
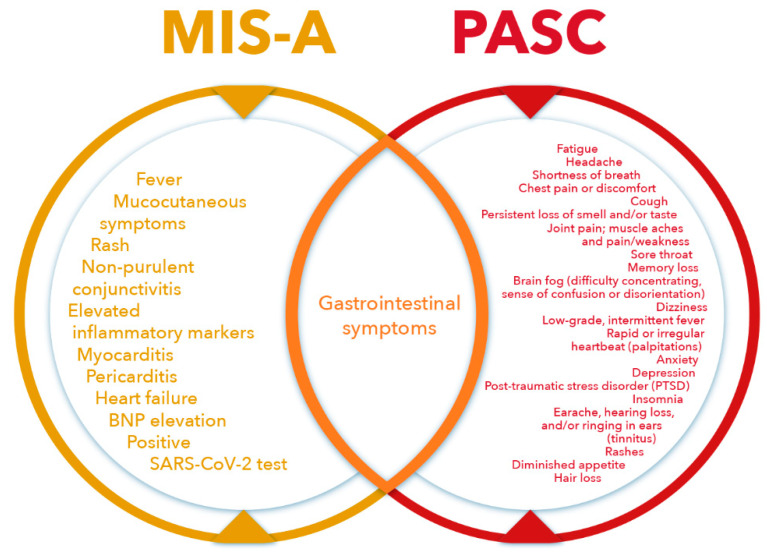
Venn diagram characterizing components in Multisystem Inflammatory Syndrome in Adults (MIS-A), Post-acute sequelae of COVID-19 (PASC), and common aspects. Abbreviations: BNP—B-type natriuretic peptide; SARS-CoV-2—Severe Acute Respiratory Syndrome Coronavirus 2.

## Data Availability

Not applicable.

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
