# Peer review of "The Multifaceted Manifestations of Multisystem Inflammatory Syndrome during the SARS-CoV-2 Pandemic"

_pathogens, 2022, doi:10.3390/pathogens11050556_

Round 1

Reviewer 1 Report

The Multifaceted Manifestations of Multisystem Inflammatory 2 Syndrome during the SARS-CoV-2 Pandemic deals with clinical manifestations and the distinguishing of KD, MIS-A,MIS-C, PASC.

1.) the main weakness of the paper is that it does not take the different strains into account and their clinical manifestations, which differ significantly. There is no indication in the text of when the different case studies occurred during the pandemic. What are the effects of Omikron on mis-c, mis-a or pasc? There are now also some publications around the topic and the review should be updated to increase its current relevance. Especially in the part about MIS-V  a timeframe is missing when these cases occurred, as it is known that vaccinated people are still vulnerable to omicron infection.

2.) English language improved in the new script, but several passages still need proofreading and polishing. 

Some examples:

Line 31-33 --> sentence order is not correct and the linking of the sentence is still poor.

Line 34: how to diagnose

Line 43-45: revise sentence 

Line 109: showed more frequently

Line 310-312: Revise sentence structure and linking. 

Line345-346: Revise

Line 392-393/ 401/406: change to indirect question 

Hyphens are missing throughout the text

3.) Add figure caption and describe figures in more detail in text. 

Author Response

The Multifaceted Manifestations of Multisystem Inflammatory 2 Syndrome during the SARS-CoV-2 Pandemic deals with clinical manifestations and the distinguishing of KD, MIS-A,MIS-C, PASC.

1.) the main weakness of the paper is that it does not take the different strains into account and their clinical manifestations, which differ significantly. There is no indication in the text of when the different case studies occurred during the pandemic. What are the effects of Omikron on mis-c, mis-a or pasc? There are now also some publications around the topic and the review should be updated to increase its current relevance. Especially in the part about MIS-V  a timeframe is missing when these cases occurred, as it is known that vaccinated people are still vulnerable to omicron infection.

2.) English language improved in the new script, but several passages still need proofreading and polishing.

Some examples:

Line 31-33 --> sentence order is not correct and the linking of the sentence is still poor.

Line 34: how to diagnose

Line 43-45: revise sentence

Line 109: showed more frequently

Line 310-312: Revise sentence structure and linking.

Line345-346: Revise

Line 392-393/ 401/406: change to indirect question

Hyphens are missing throughout the text

3.) Add figure caption and describe figures in more detail in text.

Response 1: Please provide your response for Point 1. (in red)

All strains were reviewed, including their clinical manifestations.

The effects of Omicron on MIS-C, MIS-A, and PASC are now included.

Time frame added

Response 2: Please provide your response for Point 2. (in red)

New round of proofreading and polishing.

All prepositions were edited.

Redundant, repetitive, unnecessary phrasing edited for straightforwardness.

Revisions were made to increase clarity and reduce ambiguity.

Complex structure sentences or unclear sentences are restructured.

The flow was improved.

Response 3: Please provide your response for Point 3. (in red)

Figure captions added.

Reviewer 2 Report

In this review, the authors discussed the origins of the different trajectories of the inflammatory syndromes (PASC, MIS-A, and MIS-C) related to SARS-CoV-2 infection. 

  1. It's a good literature summary and it lacks authors' perspective.
  2. Did the authors perform meta-data analysis? Any differences in cytokines or chemokines unique to the specific inflammatory syndrome?
  3. The authors need to restructure the paper for better flow to understand the concepts. 
  4. A summary of tables would be helpful. 

Author Response

Comments and Suggestions for Authors

In this review, the authors discussed the origins of the different trajectories of the inflammatory syndromes (PASC, MIS-A, and MIS-C) related to SARS-CoV-2 infection.

It's a good literature summary and it lacks authors' perspective.

Did the authors perform meta-data analysis? Any differences in cytokines or chemokines unique to the specific inflammatory syndrome?

The authors need to restructure the paper for better flow to understand the concepts.

A summary of tables would be helpful.

Response 1: Please provide your response for Point 1. (in red)

We included the differences in cytokines or chemokines.

We discuss new viral variants and clinical manifestations.

We included the effect of the Omicron variant on MIS-C, MIS-A, or PASC.

Response 2: Please provide your response for Point 2. (in red)

All prepositions were edited.

Redundant, repetitive, unnecessary phrasing edited for straightforwardness.

Revisions were made to increase clarity and reduce ambiguity.

Complex structure sentences or unclear sentences are restructured.

The flow was improved.

Reviewer 3 Report

The problem of Multisystem Inflammatory Syndrome (MIS) in COVID-19 is extremely complex, relevant and largely unresolved. The desire of the authors of the presented publication to solve this problem is commendable. The authors have presented well the clinical data, including the results of a comparative analysis on this problem. However, in general, they have so far failed to form a complete picture of the MIS pathogenesis and describe the relationship between MIS and other complications of COVID-19, even at a hypothetical level. Therefore, I suggest that the authors conceptually revise the design and content of the article and resubmit it in a corrected form.

Comments and suggestions:

  1. The purpose of the article as outlined in the Summary (24-25) “This review discusses the origins of the different trajectories of the inflammatory syndromes related to SARS-CoV-2 infection” does not correspond to the paper title: “The Multifaceted Manifestations of Multisystem Inflammatory Syndrome during the SARS- CoV-2 Pandemic. In addition, "This review discusses the origins of the different trajectories of the inflammatory syndromes related to SARS-CoV-2" is the least successful part of the paper (see below).

  1. Introduction. In this chapter, "Multisystem Inflammatory Syndrom (MIS)" is only mentioned in the last paragraph (52-54): "This review analyzes the various manifestations of infection with SARS-CoV-2.): «This review analyzes the various manifestations of infection with SARS-CoV-2. It focuses on clinical manifestations and their relationships to the diverse immune responses in multisystem inflammatory syndrome developing after or during COVID-19». Therefore, it is not at all clear what exactly the review article is about.

  1. Chapter «Coronavirus infections» (55). In this chapter, it was advisable to describe the main causes of systemic complications of COVID-19, which include MIC. Unfortunately, the authors of the article failed to do this.

  1. Chapter «Kawasaki disease (KD)» (79). The purpose of writing this chapter is unclear. It is known that KD has certain clinical and pathogenesis similarities with MIS-C in COVID-19. However, KD is not directly related to COVID-19. Apparently, the authors of the article should have first given a clear clinical and pathogenetic characterization of MIS-C/A in COVID-19, and then briefly and clearly differentiated this pathology from other complications of COVID-19 and some other diseases with similar signs, including KD.

  1. Chapter «Description of MIS-C in children» (121). This chapter is already relevant to the stated purpose of the article. It is not clear why "in children", since MIS-C can only be in children, while adults can have MIS-A. This section outlines the clinical and clinical laboratory features of MIS-C. However, the pathogenetic mechanisms of this complication, including the role of autoimmune and autoinflammatory processes, of cytokine storm and of other manifestations of systemic inflammation, the modern (data of recent months) superantigenic concept of MIS (doi:10.3390/ijms23031716), the role of genetic predisposition factors associated with HLA genes etc. are completely insufficiently.

  1. Chapter «Mast-cell activation syndrome» (337). The purpose of writing this chapter is completely incomprehensible. Idiopathic Mast Cell Activation Syndrome (MCAS) and mastocytosis are specific clinical definitions not directly related to COVID-19. At the same time, it is necessary to distinguish MCAS from the MCA phenomenon, which is one of the phenomena of systemic inflammation, including in COVID-19 (including MIS). The MCA phenomenon is associated with systemic activation of the vascular endothelium, intravascular activation of hemostasis and of complement systems. It would probably reasonable for the authors to analyze these causes of mast cell activation in MIS associated with COVID-19. The MCA phenomenon can be referred to as an MCAS-like syndrome.

  1. Chapter «Post-acute Sequelae of SARS-CoV-2 Infection» (373). PASC is a group of complications associated with long-term effects of COVID-19. The PASC problem has been described in a large number of original and review articles. The bottleneck of this problem is determining the association of PASC with a previous MIS-C/A in COVID-19. Unfortunately, the authors of the presented article failed to clarify this component of the PASC problem. Without solving this particular problem, the presented PASC data have no signs of novelty and theoretical/practical relevance.

Conclusion:

The presented paper lacks integrity and consistency. Perhaps the authors should focus on the specific problem of MIS-C/A in COVID-19 and consider it comprehensively. Other sections should be considered in relation to MIS and its manifestations such as systemic inflammation. Given the high relevance of this problem, I suggest that the authors conceptually revise the structure of the paper and present the work in a qualitatively new format.

Author Response

Comments and Suggestions for Authors

The problem of Multisystem Inflammatory Syndrome (MIS) in COVID-19 is extremely complex, relevant and largely unresolved. The desire of the authors of the presented publication to solve this problem is commendable. The authors have presented well the clinical data, including the results of a comparative analysis on this problem. However, in general, they have so far failed to form a complete picture of the MIS pathogenesis and describe the relationship between MIS and other complications of COVID-19, even at a hypothetical level. Therefore, I suggest that the authors conceptually revise the design and content of the article and resubmit it in a corrected form.

Comments and suggestions:

The purpose of the article as outlined in the Summary (24-25) “This review discusses the origins of the different trajectories of the inflammatory syndromes related to SARS-CoV-2 infection” does not correspond to the paper title: “The Multifaceted Manifestations of Multisystem Inflammatory Syndrome during the SARS- CoV-2 Pandemic. In addition, "This review discusses the origins of the different trajectories of the inflammatory syndromes related to SARS-CoV-2" is the least successful part of the paper (see below).

Introduction. In this chapter, "Multisystem Inflammatory Syndrom (MIS)" is only mentioned in the last paragraph (52-54): "This review analyzes the various manifestations of infection with SARS-CoV-2.): «This review analyzes the various manifestations of infection with SARS-CoV-2. It focuses on clinical manifestations and their relationships to the diverse immune responses in multisystem inflammatory syndrome developing after or during COVID-19». Therefore, it is not at all clear what exactly the review article is about.

Chapter «Coronavirus infections» (55). In this chapter, it was advisable to describe the main causes of systemic complications of COVID-19, which include MIC. Unfortunately, the authors of the article failed to do this.

Chapter «Kawasaki disease (KD)» (79). The purpose of writing this chapter is unclear. It is known that KD has certain clinical and pathogenesis similarities with MIS-C in COVID-19. However, KD is not directly related to COVID-19. Apparently, the authors of the article should have first given a clear clinical and pathogenetic characterization of MIS-C/A in COVID-19, and then briefly and clearly differentiated this pathology from other complications of COVID-19 and some other diseases with similar signs, including KD.

Chapter «Description of MIS-C in children» (121). This chapter is already relevant to the stated purpose of the article. It is not clear why "in children", since MIS-C can only be in children, while adults can have MIS-A. This section outlines the clinical and clinical laboratory features of MIS-C. However, the pathogenetic mechanisms of this complication, including the role of autoimmune and autoinflammatory processes, of cytokine storm and of other manifestations of systemic inflammation, the modern (data of recent months) superantigenic concept of MIS (doi:10.3390/ijms23031716), the role of genetic predisposition factors associated with HLA genes etc. are completely insufficiently.

Chapter «Mast-cell activation syndrome» (337). The purpose of writing this chapter is completely incomprehensible. Idiopathic Mast Cell Activation Syndrome (MCAS) and mastocytosis are specific clinical definitions not directly related to COVID-19. At the same time, it is necessary to distinguish MCAS from the MCA phenomenon, which is one of the phenomena of systemic inflammation, including in COVID-19 (including MIS). The MCA phenomenon is associated with systemic activation of the vascular endothelium, intravascular activation of hemostasis and of complement systems. It would probably reasonable for the authors to analyze these causes of mast cell activation in MIS associated with COVID-19. The MCA phenomenon can be referred to as an MCAS-like syndrome.

Chapter «Post-acute Sequelae of SARS-CoV-2 Infection» (373). PASC is a group of complications associated with long-term effects of COVID-19. The PASC problem has been described in a large number of original and review articles. The bottleneck of this problem is determining the association of PASC with a previous MIS-C/A in COVID-19. Unfortunately, the authors of the presented article failed to clarify this component of the PASC problem. Without solving this particular problem, the presented PASC data have no signs of novelty and theoretical/practical relevance.

Conclusion:

The presented paper lacks integrity and consistency. Perhaps the authors should focus on the specific problem of MIS-C/A in COVID-19 and consider it comprehensively. Other sections should be considered in relation to MIS and its manifestations such as systemic inflammation. Given the high relevance of this problem, I suggest that the authors conceptually revise the structure of the paper and present the work in a qualitatively new format.

Response 1: Please provide your response for Point 1. (in red)

Revisions were made to increase clarity and reduce ambiguity.

Complex structure sentences or unclear sentences are restructured.

The flow was improved.

Response 2: Please provide your response for Point 2. (in red)

Now included.

Redundant, repetitive, unnecessary phrasing edited for straightforwardness.

The flow was improved.

Response 3: Please provide your response for Point 3. (in red)

We removed the KD section; included pertinent data in the MIS-C section.

Response 4: Please provide your response for Point 4. (in red)

Pathogenetic mechanisms new data added.

Response 5: Please provide your response for Point 5. (in red)

We removed Mast-Cell Activation Syndrome; included pertinent data in the pathogenesis discussion.

Response 6: Please provide your response for Point 6. (in red)

New data added.

Response 7: Please provide your response for Point 7. (in red)

All prepositions were edited.

Redundant, repetitive, unnecessary phrasing edited for straightforwardness.

Revisions were made to increase clarity and reduce ambiguity.

Complex structure sentences or unclear sentences are restructured.

The flow was improved.

Round 2

Reviewer 2 Report

The authors improved the manuscript based on the reviewers comments. 

Reviewer 3 Report

The authors presented an improved version of the article. All comments and suggestions were taken into account. The work can be accepted for publication in its present form.